# Research on Lightweight Disaster Classification Based on High-Resolution Remote Sensing Images

**Jianye Yuan [1], Xin Ma [2],*, Ge Han [3], Song Li [1] and Wei Gong [1,4]**

1   School of Electronic Information, Wuhan University, Wuhan 473072, China; yuan666@whu.edu.cn (J.Y.); ls@whu.edu.cn (S.L.); weigong@whu.edu.cn (W.G.)
2   State Key Laboratory of Information Engineering in Surveying, Mapping and Remote Sensing, Wuhan University, Wuhan 430079, China
3   School of Remote Sensing and Information Engineering, Wuhan University, Wuhan 430079, China; udhan@whu.edu.cn
4   Hubei Luojia Laboratory, Wuhan 430079, China
*   Correspondence: maxinwhu@whu.edu.cn

**Abstract:** With the increasing frequency of natural disasters becoming, it is very important to classify and identify disasters. We propose a lightweight disaster classification model, which has lower computation and parameter quantities and a higher accuracy than other classification models. For this purpose, this paper specially proposes the SDS-Network algorithm, which is optimized on ResNet, to deal with the above problems of remote sensing images. First, it implements the spatial attention mechanism to improve the accuracy of the algorithm; then, the depth separable convolution is introduced to reduce the number of model calculations and parameters while ensuring the accuracy of the algorithm; finally, the effect of the model is increased by adjusting some hyperparameters. The experimental results show that, compared with the classic AlexNet, ResNet18, VGG16, VGG19, and Densenet121 classification models, the SDS-Network algorithm in this paper has a higher accuracy, and when compared with the lightweight models mobilenet series, shufflenet series, squeezenet series, and mnasnet series, it has lower model complexity and a higher accuracy rate. According to a comprehensive performance comparison of the charts made in this article, it is found that the SDS-Network algorithm is still better than the regnet series algorithm. Furthermore, after verification with a public data set, the SDS-Network algorithm in this paper is found to have a good generalization ability. Thus, we can conclude that the SDS-Network classification model of the algorithm in this paper has a good classification effect, and it is suitable for disaster classification tasks. Finally, it is verified on public data sets that the proposed SDS-Network has good generalization ability and portability.

**Keywords:** disaster classification; attention mechanism; depth separable; calculation amount; parameter amount; Grad-CAM

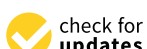

## 1. Introduction

In recent years, due to the continuous increase in global greenhouse gas emissions, there have been increasingly more problems, such as melting glaciers and climate warming, and natural disasters have also continued to increase. Natural disasters are mainly classified into four categories: meteorological disasters, geological disasters, biological disasters, and astronomical disasters [1]. This article classifies disasters into car accidents, floods, fires, hurricanes, and earthquakes. Traditionally, these disasters cannot be classified or identified automatically, and they can only be prevented and processed in specific disaster areas [2,3].

In view of the continuous development of remote sensing images, deep learning methods have become increasingly more important in the classification of disasters. For example, Ahmed Ahmouda [4] et al. mapped short- and long-term changes in behavior and tweeting activity in areas affected by natural disasters by analyzing earthquakes in

Nepal and central Italy. Scientists have used deep learning technology to study information data, land coverage, floods, etc., in disaster areas, and they found that disasters still have a high research value [5–8].

Many researchers have proposed their own disaster classification methods. For example, Cheng Ximeng of the China University of Geosciences (Beijing) [9] automatically classified disasters based on high-resolution remote sensing images in combination with earthquake disasters and proposed a rapid earthquake disaster assessment model. To verify the effectiveness of the model, Liu Hongyan [10] et al. classified sudden geological disasters into four categories in their study, and they proposed an improved monitoring and early warning and prediction method of movement distance after instability for emergency prevention. Xu Anxin of Shandong University [11] used SVM to propose a power grid meteorological disaster early warning method based on scene classification and recognition. This method can better extract the meteorological disaster category, and identify and predict power grid faults more accurately to improve the outcome of power grid meteorological disasters, which lays the foundation for improving the warning ability for power grid meteorological disasters. The above methods are all used to classify and recognize a specific disaster, and they have higher requirements for specific scenarios. In view of this, this paper proposes a lightweight disaster classification model (Spatial Depthwise Separable Convolution, SDS-Network) using high-resolution remote sensing images, which may further improve the accuracy rate of disaster classification and reduce the calculations and parameters of the algorithm.

### 1.1. Remote Sensing Images

Deep learning technology has developed rapidly in recent years, and it is increasingly being combined with remote sensing images. Below is an introduction to remote sensing images and deep-learning-related knowledge.

Remote sensing images are generally obtained from top-to-bottom image information captured by airborne or spaceborne equipment, satellites, and other tools. In traditional remote sensing image classification tasks, the minimum distance method [12], parallelepiped method [13], maximum likelihood method [14], and other methods are more commonly used due to their foreign matters being in the same spectrum, as well as other characteristics. Therefore, the accuracy of classification needs to be further improved. With the development of remote sensing technology, Liu Jiajia [15] et al. elaborated on the classification of urban buildings based on remote sensing images; Li Anqi et al. [16] designed a typical crop classification method based on the U-Net algorithm; and Wang Ziqi [17] et al. adopted a knowledge map to supplement the classification of remote sensing positioning, which reduced the image retrieval time by half. Better ideas for the classification and positioning of remote sensing images have been put forward in the above methods, but they all have specific application scenarios, which are limited in the classification of high-resolution remote sensing images in the disaster classification scenario in this paper. In view of this, this paper proposes the SDS-Network, which can be used in disaster classification tasks.

### 1.2. Deep Learning

The deep learning method was developed around 2000, and it can better ascertain useful information in original images and process correlations. To date, it has been effectively used in target detection [18], natural language processing [19], speech processing, [20], and semantic analysis [21], and it has greatly promoted the development of artificial intelligence. Moreover, it is mainly integrated into the multi-layer perceptron model [22], deep neural network model [23], and recurrent neural network model [24], including many representatives, such as the deep belief network (DBN), convolution neural network (CNN), and recurrent neural network (RNN). CNN models, including LeNet5 [25], AlexNet [26], VGG [27], GoogleNet [28], ResNet [29], Wide ResNet [30], Xception [31], DenseNet [32], SEnet [33], squeeze [34], MobileNet [35], and Shuffle [36], are mainly used in image classification. Among them, the jump connection of ResNet makes a great contribution, and it

solves gradient disappearance and explosion with the deepening of the model [29]; squeeze, Shuffle, and MobileNet are lightweight models, and they solve the problem of operation models on embedded devices such as mobile phones. With the continuous development of CNN, increasingly more classification models are being proposed by research scholars to further meet the needs of life and industrial production. The algorithm proposed in this article makes its own contribution to the scientific community.

## 2. Related Work

This article studied various classification models and, at the same time, optimized the models used in the classification of high-resolution remote sensing images. The classification model selected in this article was optimized on ResNet50. Finally, ResNet50 was optimized using the spatial attention mechanism, depthwise separable convolution, and hyperparameter tuning.

### 2.1. Spatial Attention Mechanism

The spatial attention mechanism weighs the spatial information in the spatial dimension. Its working principle is as follows: Firstly, create a feature map $F'$ of H * H * C size (H represents the length or width of the feature map, and C represents the number of channels of the feature map), and then use the maximum pooling and average pooling to reorganize the feature maps and obtain two Conv-x feature information descriptions with a size of H * H * 1. Next, concatenate these Conv-x feature information descriptions according to channel correlation. After splicing is completed, use a 7 * 7 convolutional layer Conv-y and the activation function (Sigmoid) [37] to obtain the weight coefficient M on the spatial dimension of the feature map, and the feature map is Conv. The convolution operation part of the spatial attention mechanism proposed in the algorithm in this paper is shown in Figure 1. Finally, multiply the weight coefficient M and the input feature map information $F'$ into the output feature map of the spatial attention mechanism, as shown in Formula (1).

$$
\begin{aligned}
M \quad &= \sigma\left(f^{7*7}([AvgPool(F), MaxPool(F)])\right) \\
&= \sigma(f^{7*7}([F_{avg}^S; F_{max}^S \\
M' \quad &= M * F'
\end{aligned}
\tag{1}
$$

where *AvgPool* represents the average pooling; *MaxPool* represents the maximum pooling; $\sigma$ represents the Sigmoid activation function; and $M'$ represents the result of multiplying the weight coefficient M and the input feature map $F'$. In Figure 1, U is the input feature map; V is the output feature map; and the remainder represent the conversion modules of the spatial attention mechanism.

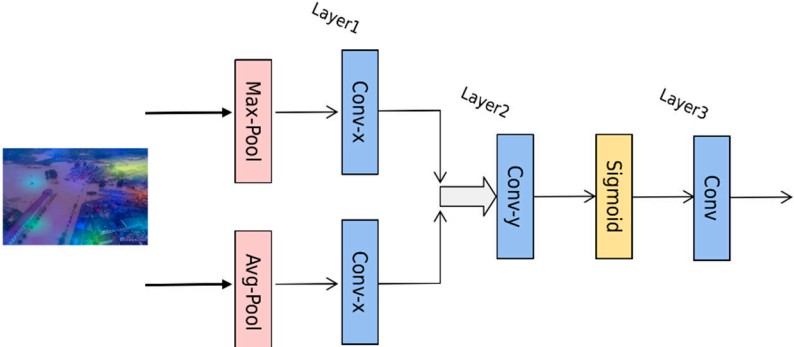

**Figure 1.** Algorithm structure diagram of spatial attention mechanism.

### 2.2. Depthwise Separable Convolution

In order to reduce the calculations and parameters of the algorithm, this paper continues to optimize the algorithm model, and it proposes depthwise separable convolution (DSC) [38], which introduces the 3 * 3 convolution of ResNet50 into depthwise convolution.

The depth separable convolution is composed of depthwise convolution (DC) [39] and pointwise convolution (PC) [40]. The calculation methods of standard convolution (SC) and DC are shown in Formulas (2)–(5). It can be seen in Formula (6) that the calculations of standard convolution (SC) are much greater than those of DSC.

Assuming that the size of the input feature map is $D_k * D_k * M$, that the size of the convolution kernel is $D_f * D_f * M$ (the number of which is N), and that each point in the corresponding space position of the feature map will perform a convolution operation, then it can be seen that a single convolution requires $D_k * D_k * D_f * D_f * M$ calculations. Therefore, for a single convolution, the calculations are as follows:

$$\text{SC FLOPs} \qquad\qquad SC = D_k * D_k * D_f * D_f * M * N \qquad\qquad (2)$$

$$\text{DC} \qquad\qquad = D_k * D_k * D_f * D_f * M \qquad\qquad (3)$$

$$\text{PC} \qquad\qquad = M * N * D_k * D_k \qquad\qquad (4)$$

$$\text{DSC FLOPs} \qquad DSC = DC + PC = D_k * D_k * D_f * D_f * M + M * N * D_k * D_k \qquad (5)$$

The ratio of calculations of DSC to ordinary convolution is FLOPs:

$$\text{FLOPs} = \frac{DSC}{SC} = \frac{Dk * Dk * Df * Df * M + M * N * Dk * Dk}{Dk * Dk * Df * Df * M * N} = \frac{1}{N} + \frac{1}{D_f^2} \qquad (6)$$

Standard convolution means that the convolution is performed on each feature channel, but it can be seen in Figure 2 that the depth separable convolution performs convolution on each channel, so the calculations and parameters of the model greatly reduce, which increases the efficiency of the algorithm. This paper adds depth separable convolution to ResNet's Bottleneck's 3 * 3 convolution blocks, thereby reducing the calculations and parameters of the disaster classification model and improving the operating efficiency of the model.

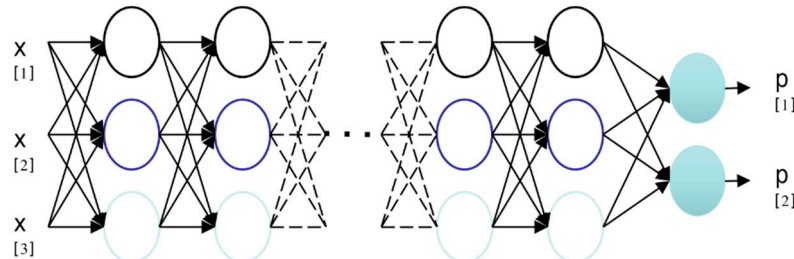

**Figure 2.** SC operation diagram.

To illustrate this, Figures 2 and 3 use three nodes for input and two nodes for output. Figure 2 presents an operation diagram of the standard convolution. When the input values $X_{[1]}$, $X_{[2]}$, and $X_{[3]}$ are connected to the neuron, the neuron performs the calculation with each neuron in the next layer, and finally, the number of output categories is obtained. Figure 3 shows the mechanism of deep separable convolution, where the calculation is only performed with neurons in the same layer and not in the convolution of other layers. The calculations and parameters of the algorithm should reduce, and the operation speed will improve.

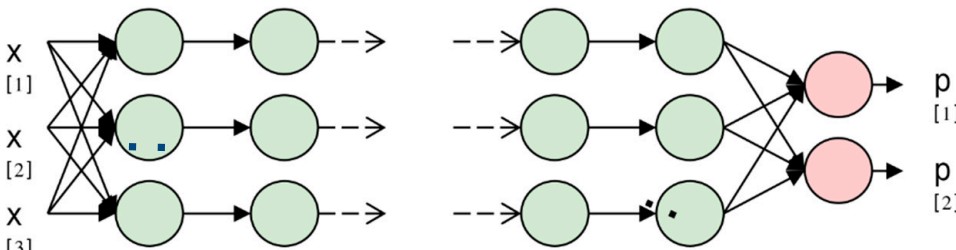

**Figure 3.** DSC operation diagram.

### 2.3. Hyperparameter Adjustment

In order to obtain better experimental results, the hyperparameters adjusted this time are mainly in the model structure and training links. This paper first reduced the residual blocks of ResNet and reduced the original 16-layer residual block of [3,4,6,3] into a 4-layer residual block; after that, the parameters in the training process were adjusted, and the optimizer SGD [40] before training was replaced with AdamW [41]. In addition, the initial learning rate was set to 0.01–0.03, and the weight decay was set to 0.0004; finally, this article replaced the activation function ReLU [42] with the GELU activation function [43]. The structure of each spatial depthwise separable block (SDS-Block) in this article after optimization is shown in Table 1. From the table, it can be concluded that the number of convolutional layers for each SDS-Block is 5.

**Table 1.** SDS-Block model structure.

| | Model | | Convolution Kernel and Parameters |
|---|---|---|---|
| | ConV1 | | 1 * 1, Stride = 1 |
| | BN1 | | |
| | ConV2 | | 3 * 3, Stride = 1, padding = 1, Group |
| | BN2 | | |
| | ConV3 | | 1 * 1, Stride = 1, padding = 1 |
| | BN3 | | |
| | GELU | | |
| | | ConV1 | 7 * 7, Stride = 1, padding = 3 |
| | | Sigmoid | |
| Spatial attention | | ConV2d | 1 * 1, Stride = 1 |
| | | BN2d | |

### 2.4. Algorithm Structure

In summary, in the design of the disaster classification model in this paper, the spatial attention mechanism, depthwise separable convolution, and hyperparameter adjustment are used to improve the accuracy of the algorithm and reduce calculations and parameters. As shown in Table 2, the Spatial Depthwise Separable Network (SDS-Network) algorithm model is designed for this article. As can be seen in Table 2, the algorithm in this paper is composed of convolution, an activation function, a pooling layer, an SDS-Block module, and a full-face hierarchy, and finally, it constitutes a new convolutional neural network model for disaster classification. Since each SDS-Block is communicated through 4 layers of convolutional layers, the SDS-Network in this article only has 16 layers, which is in line with the structure of lightweight models.

**Table 2.** Structure of algorithm model.

| Model | Output Size | Kernel/Parameters | FLOPs | Parameters |
|---|---|---|---|---|
| Conv1 | 112 * 112 | 7 * 7, Stride = 2, Padding = 3 | 118,013,952 | 9408 |
| BN1 | 112 * 112 | | 1,605,632 | 128 |
| GELU | 112 * 112 | | 0 | 0 |
| Max pool | 56 * 56 | 3 * 3, Stride = 2, Padding = 1 | 802,816 | 0 |
| SDS-Block | 56 * 56 | SDS-Block-1 | 121,733,248 | 38,818 |
| SDS-Block | 28 * 28 | SDS-Block-2 | 260,490,272 | 233,186 |
| SDS-Block | 14 * 28 | SDS-Block-3 | 258,676,488 | 925,026 |
| SDS-Block | 7 * 7 | SDS-Block-4 | 412,180,307 | 3,694,962 |
| adaptAvgPool | 1 * 1 | | 0 | 0 |
| Linear | 5 | | 10,240 | 10,245 |
| **Total** | | | **1,019,116,650** | **4,901,773** |

## 3. Experimental Link

### 3.1. Experimental Setup

This experiment was carried out on an Ubuntu 20.04 system with RTX 3090 (24 G of video memory), adopted a Pytorch 1.8 operation framework, and selected SGD as the pre-optimizer. The initial learning rate was 0.01, and the learning rate for each epoch was 95% of the original. Random seeds were used to input pictures to ensure the stability of the experimental data. The training set and test set in this article are from the Kaggle Disaster Classification Competition and the internal data set of the State Key Laboratory of Information Engineering in Surveying, Mapping and Remote Sensing, Wuhan University. There are 11,243 data sets in this experiment, comprising 8997 training sets and 2246 test sets, each of which is a color image. However, in order to increase the robustness of model training, the length, width, and size of the images are not limited. When the models were input, the images were randomly cropped to a size of 224*224, and data enhancement was used to further improve the data.

### 3.2. Experimental Procedure

This paper first used the spatial attention mechanism module to improve the accuracy of the algorithm, and then it introduced deep separable convolution to reduce the calculations and parameters of the algorithm while ensuring accuracy; finally, the hyperparameters were fine-tuned to optimize the model and obtain the SDS-Network algorithm model in this paper. The evaluation indicators used in this article include floating point operations (FLOPs), parameters, accuracy (Acc top1), and memory.

In Table 3, ResNet50 + Spatial represents the use of the spatial attention mechanism on ResNet50; ResNet50 + Spatial + DepthWise represents the further use of the depthwise separable convolution; and SDS-Network is the algorithm model proposed in this paper. It can be seen in Table 3 that the Acc of the original model ResNet50 was 0.8998 and that the optimized Acc was 0.9248, which increased by 2.5%; the parameters were reduced by about 6 times compared with those of ResNet50, FLOPs were reduced by about 4 times, and memory was reduced 2 times. Therefore, the SDS-Network algorithm proposed in this paper has achieved good results and is suitable for disaster classification.

**Table 3.** Comparison of data performance between SDS-Network and ResNet50.

| Model | Acc (Top1) | Parameters | FLOPs (G) | Memory (M) |
|---|---|---|---|---|
| ResNet50 [24] | 0.8998 | 25,557,032 | 4.12 | 109.69 |
| ResNet50 + Spatial | 0.9110 | 25,558,600 | 4.12 | 109.79 |
| ResNet50 + Spatial + DepthWise | 0.9118 | 14,275,336 | 2.28 | 109.79 |
| **SDS-Network** | **0.9248** | **4,901,773** | **1.02** | **48.47** |

As can be seen in Figure 4, the SDS-Network classification algorithm has an Acc of 97% for Accident, 86% for Cyclone, 92% for Earthquake, 95% for Flood, and 87% for Wildfire, which are all greater than 85%. It can also be seen that the Acc of Accident is 0.11 higher than that of Cyclone, with a smaller difference. Because the SDS-Network is applicable to each category of disaster classification, there is no notable difference.

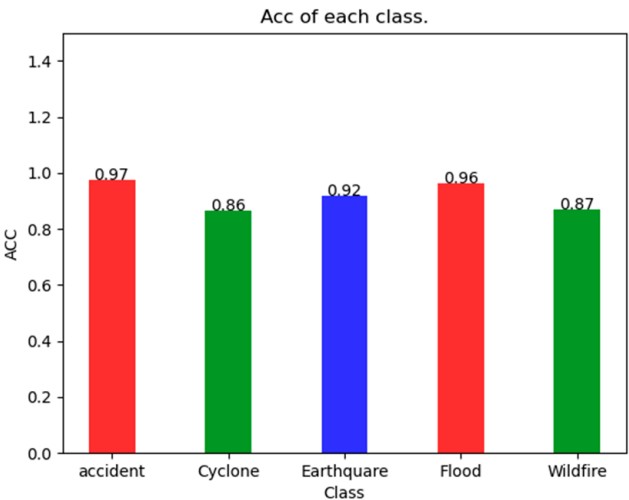

**Figure 4.** Acc of SDS-Network algorithm in each category.

It can be seen in Figures 5 and 6 that, as the iterations of the SDS-Network algorithm increased, the Acc also increased, and Loss decreased. In the graph, it can be seen that the trained Acc was slightly higher than that in the testing phase, and the trained Loss was slightly lower than that in the testing phase. In order to ensure practicability and applicability, the Acc values selected in this article were all from the test phase. When the epoch was iterated 100 times, both Acc and Loss tended to be stable, indicating that setting the epoch to 100 in this article was consistent with the experimental environment.

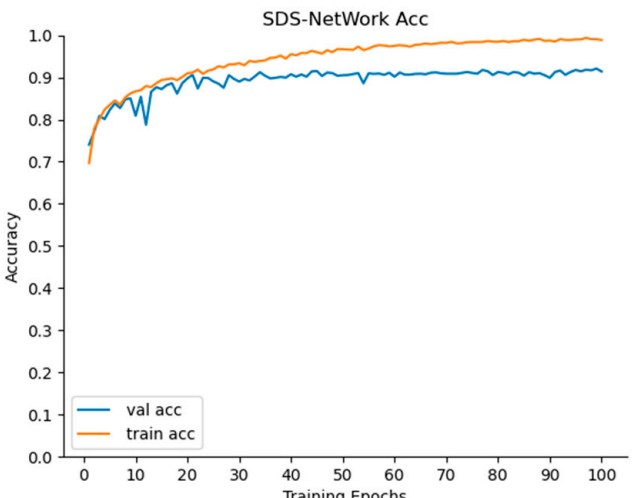

**Figure 5.** Acc curve of accuracy rate of SDS-Network algorithm.

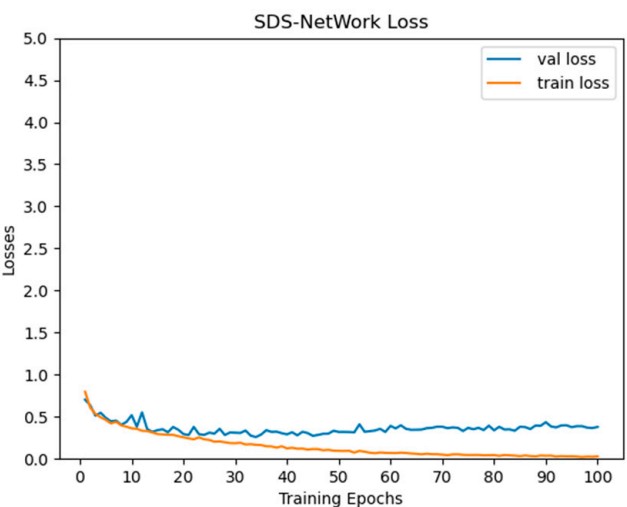

**Figure 6.** Loss curve diagram of SDS-Network algorithm.

This article first compared the SDS-Network with some classic classification algorithms. It can be seen in Table 4 that the Acc of Densenet121 was higher than that of AlexNet and ResNet18, but the Acc of the SDS-Network was higher than that of Densenet121, and the Acc of the SDS-Network algorithm was lower than that of AlexNet, ResNet18, and Densenet121. Although AlexNet was better than the SDS-Network algorithm in FLOPs and memory, its Acc was lower, and Vgg16 and Vgg19 performed poorly in Acc testing. Therefore, the SDS-Network algorithm proposed in this paper performed best in disaster classification.

**Table 4.** Comparison of SDS-Network model and classic algorithms.

| Model | Acc (Top1) | Parameters | FLOPs | Memory (M) |
|---|---|---|---|---|
| AlexNet [26] | 0.8740 | 61,100,840 | 715.54 M | 4.19 MB |
| ResNet18 [29] | 0.9092 | 11,689,512 | 1.82 G | 25.65 MB |
| Vgg16 [27] | 0.0824 | 138,357,544 | 15.5 G | 109 MB |
| Vgg19 [27] | 0.0334 | 143,667,240 | 19.67 G | 119.34 MB |
| Densenet121 [32] | 0.9132 | 6,958,981 | 2.88 GFlops | 147.10 MB |
| **SDS-Network** | **0.9248** | **4,901,773** | **1.02 G** | **48.47 MB** |

In order to further verify the lightweight performance of the SDS-Network algorithm, this paper conducted further experimental research and compared the SDS-Network algorithm with the lightweight algorithms mobilenet series, shufflenet series, squeezenet series, and mnasnet series [44].

First of all, in the mobilenet series, mobilenet_v3_large had the best overall performance. However, the FLOPs in mobilenet_v3_large were slightly higher than those of the SDS-Network algorithm, and mobilenet_v3_large performed slightly lower than the SDS-Network algorithm in Acc, parameters, and memory. After a comprehensive evaluation, the SDS-Network algorithm was found to be slightly better than the mobilenet_V3_large algorithm in lightweight performance. Secondly, among the shufflenet series of algorithms, shufflenet_v2_x1_0 had the best overall performance. Its performance was better than the SDS-Network algorithm in parameters, FLOPs, and memory, but its Acc was slightly lower than that of the SDS-Network algorithm. We can see in Figure 7 that the SDS-Network algorithm was more stable than the shufflenet_V2_X1_0 algorithm in each Acc value when comparing the effects of each category. Therefore, it is concluded that both the SDS-Network and shufflenet_v2_x1_0 algorithms are more suitable for lightweight disaster classifications. Finally, Table 5 proves that the squeezenet and mnasnet series are very low in Acc and that the algorithm model has an overfitting problem, so it is not suitable for disaster classifications.

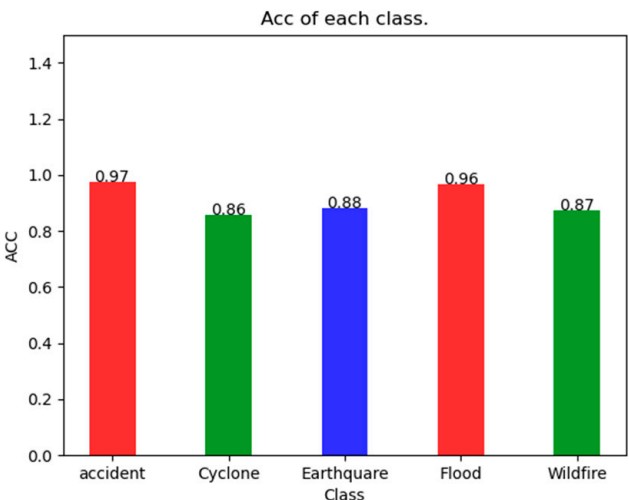

**Figure 7.** Acc effect diagram of each type of shufflenet_v2_x1_0 algorithm.

**Table 5.** Comparison of SDS-Network model and lightweight algorithm.

| Model | Acc (Top1) | Parameters | FLOPs | Memory (M) |
|---|---|---|---|---|
| mobilenet_v2 [30] | 0.9038 | 3,504,872 | 0.31 | 74.25 |
| mobilenet_v3_small [30] | 0.8967 | 2,542,856 | 0.06 | 16.20 |
| mobilenet_v3_large [30] | 0.9061 | 5,483,032 | 0.22 | 50.40 |
| shufflenet_v2_x0_5 [31] | 0.9025 | 1,366,792 | 0.04 | 11.24 |
| shufflenet_v2_x1_0 [31] | 0.9127 | 2,278,604 | 0.15 | 20.85 |
| shufflenet_v2_x1_5 [31] | 0.9096 | 3,503,624 | 0.30 | 29.32 |
| shufflenet_v2_x2_0 [31] | 0.9127 | 7,393,996 | 0.58 | 39.51 |
| squeezenet1_0 [29] | *0.5913* | 1,248,424 | 0.81 | 35.60 |
| squeezenet1_1 [29] | *0.5690* | 1,235,496 | 0.35 | 21.35 |
| mnasnet0_5 [29] | *0.3664* | 2,218,512 | 0.11 | 33.65 |
| mnasnet0_75 [29] | *0.3339* | 1,235,496 | 0.22 | 52.22 |
| mnasnet1_0 [39] | *0.3339* | 4,383,312 | 0.32 | 59.94 |
| mnasnet1_3 [39] | *0.3339* | 6,282,256 | 0.53 | 80.66 |
| **SDS-Network** | **0.9248** | **4,901,773** | **1.02** | **48.47** |

Next, the algorithm proposed in this paper was further compared with the algorithms in the efficient series [45], which are relatively new image classification models. In Table 6, it can be seen that the SDS-Network still maintains the highest Acc while also maintaining the best effect on memory, FLOPs, and parameters (state of the art). Although the parameters of efficientnet_b0 were slightly lower than those of the SDS-Network algorithm, its comprehensive performance was significantly worse than that of the SDS-Network algorithm. Therefore, the SDS-Network algorithm is better than the efficient series of algorithms.

**Table 6.** Comparison of SDS-Network model and efficient series of algorithms.

| Model | Parameters | FLOPs (G) | Memory (M) | Acc (Top1) |
|---|---|---|---|---|
| efficientnet_b0 [40] | 4,015,234 | 0.39 | 79.40 | 0.9176 |
| efficientnet_b1 [40] | 6,520,870 | 0.57 | 110.64 | 0.9167 |
| efficientnet_b2 [40] | 7,709,448 | 0.66 | 115.96 | 0.9172 |
| efficientnet_b3 [40] | 10,703,917 | 0.96 | 153.93 | 0.9185 |
| efficientnet_b4 [40] | 17,559,374 | 1.54 | 201.26 | 0.9092 |
| efficientnet_b5 [40] | 28,353,078 | 2.4 | 277.07 | 0.9114 |
| efficientnet_b6 [40] | 40,749,534 | 3.42 | 354.58 | 0.9038 |
| efficientnet_b7 [40] | 63,802,326 | 5.25 | 474.66 | 0.8972 |
| **SDS-Network** | **4,901,773** | **1.02** | **48.47** | **0.9248** |

In order to ensure the effectiveness of the algorithm, this paper continued to compare the data with those of the regnet series [46] algorithm, and it conducted experiments in the same experimental environment. In order to obtain a more intuitive comparison, this paper drew a bar chart according to the algorithm data in Table 7, as shown in Figures 8–11. In order to facilitate understanding, in the figures, regnetx_200mf is abbreviated as x_200mf and the SDS-Network is abbreviated as SDS.

**Table 7.** Comparison of SDS-Network model and regnet series of algorithms.

| Model | Acc (Top1) | Parameters | FLOPs (G) | Memory (M) |
| --- | --- | --- | --- | --- |
| regnetx_200mf [41] | 0.9007 | 2,318,006 | 0.20 | 23.53 |
| regnetx_400mf [41] | 0.8989 | 4,774,822 | 0.39 | 36.37 |
| regnetx_600mf [41] | 0.9007 | 5,670,214 | 0.59 | 45.54 |
| regnetx_800mf [41] | 0.9007 | 6,590,694 | 0.79 | 59.65 |
| regnetx_1.6gf [41] | 0.8998 | 8,282,614 | 1.62 | 96.32 |
| regnetx_3.2gf [41] | 0.8998 | 14,293,606 | 3.2 | 141.74 |
| regnetx_4.0gf [41] | 0.9065 | 20,765,414 | 3.99 | 152.71 |
| regnetx_6.4gf [41] | 0.9065 | 24,594,006 | 6.5 | 199.57 |
| regnetx_8.0gf [41] | 0.9029 | 37,663,174 | 8.03 | 177.32 |
| regnetx_12gf [41] | 0.9025 | 43,878,502 | 12.13 | 262.67 |
| regnetx_16gf [41] | 0.9025 | 52,241,830 | 16.0 | 317.08 |
| regnetx_32gf [41] | 0.8994 | 105,305,686 | 31.82 | 455.66 |
| regnety_200mf [41] | 0.8967 | 2,796,210 | 0.20 | 22.17 |
| regnety_400mf [41] | 0.9003 | 3,905,790 | 0.40 | 41.51 |
| regnety_600mf [41] | 0.9065 | 5,449,814 | 0.60 | 46.67 |
| regnety_800mf [41] | 0.9034 | 5,498,782 | 0.79 | 56.66 |
| regnety_1.6gf [41] | 0.9069 | 10,318,764 | 1.63 | 93.16 |
| regnety_3.2gf [41] | 0.9034 | 17,932,416 | 3.2 | 129.56 |
| regnety_4.0gf [41] | 0.9025 | 19,564,190 | 4.0 | 139.98 |
| regnety_6.4gf [41] | 0.9087 | 29,294,034 | 6.39 | 189.38 |
| regnety_8.0gf [41] | 0.9092 | 37,175,170 | 8.0 | 203.30 |
| regnety_12gf [41] | 0.8976 | 49,594,990 | 12.14 | 242.96 |
| regnety_16gf [41] | 0.9078 | 80,583,290 | 15.96 | 262.04 |
| regnety_32gf [41] | 0.9074 | 141,356,048 | 32.34 | 348.16 |
| **SDS-Network** | **0.9248** | **4,901,773** | **1.02** | **48.47** |

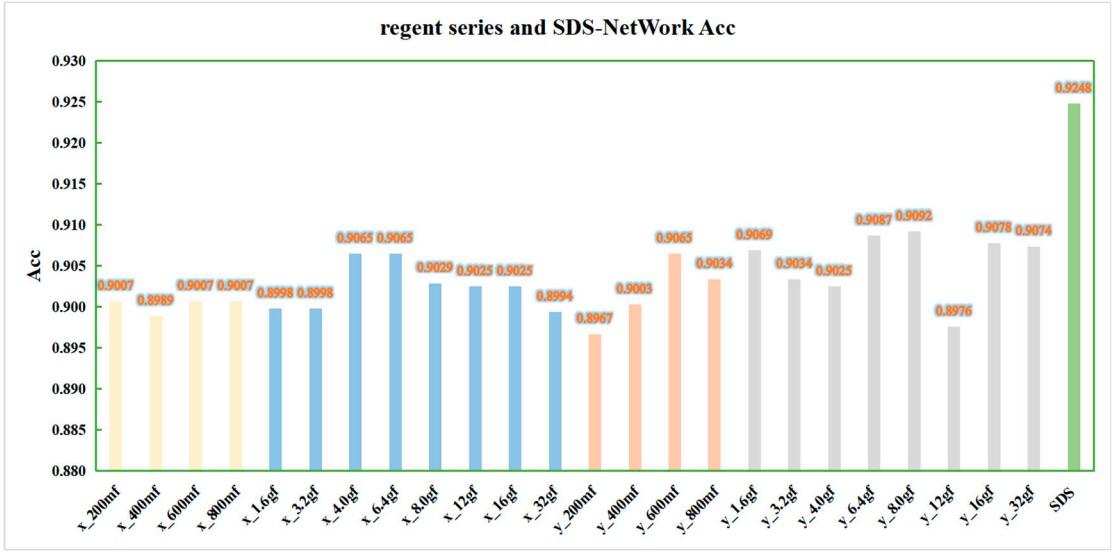

**Figure 8.** SDS-Network and regnet algorithm Acc comparison diagram. x_200mf–x_800mf are marked in yellow; x_1.6gf–x_32gf are marked in blue; y_200mf–y_800mf are marked in orange; y_1.6gf–y_32gf are marked in gray; and SDS is marked in green.

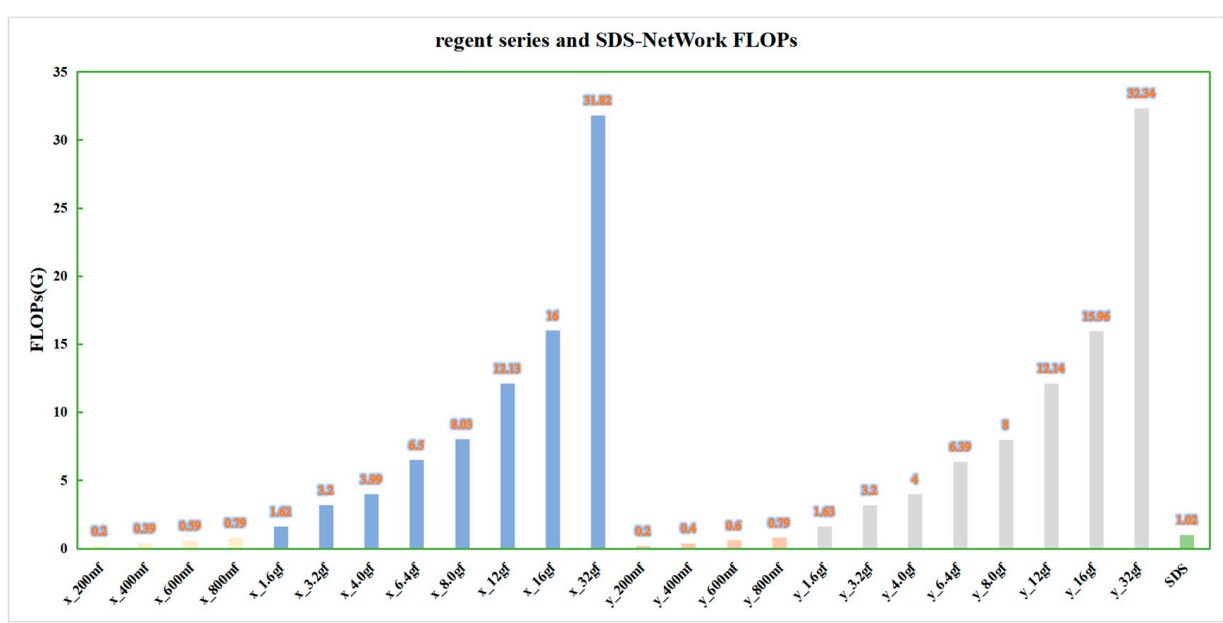

**Figure 9.** Comparison of FLOPs between SDS-Network and regnet algorithms.

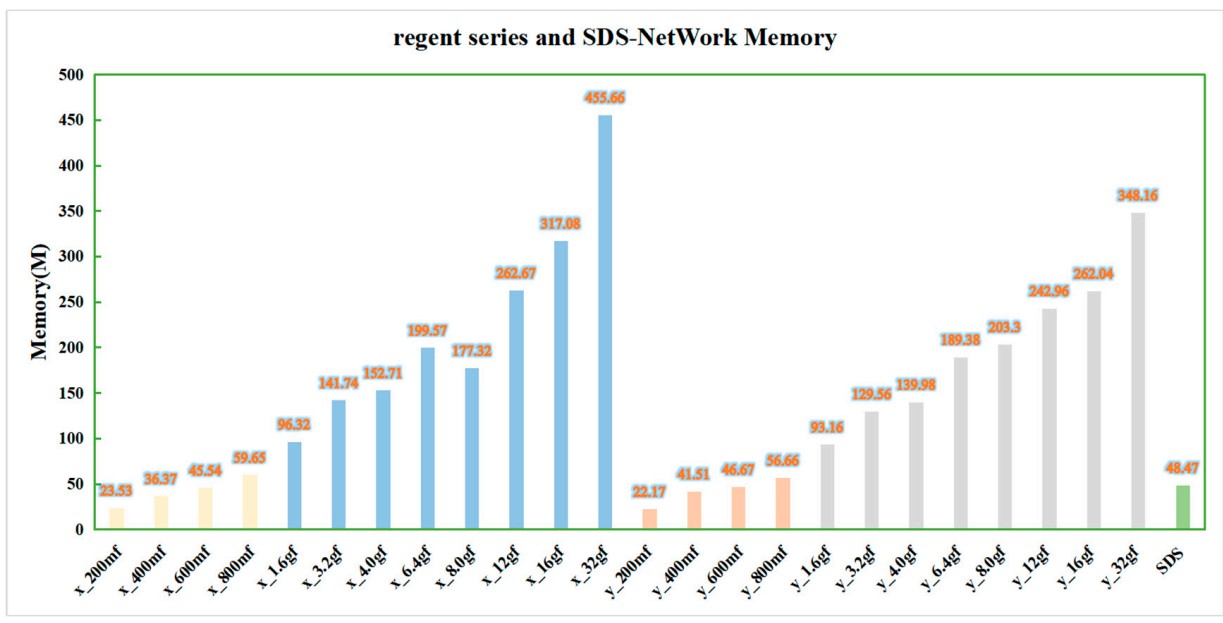

**Figure 10.** Memory comparison diagram of SDS-Network and regnet algorithm.

It can be seen in Figure 8 that the data of the regnetx_y series of algorithms are slightly higher quality than those of the regnetx_x series of algorithms, but they are all lower quality than those of the SDS-Network algorithm. Therefore, the SDS-Network algorithm performs better in disaster classifications.

It can be seen in Figure 9 that the regnetx_32gf and regnety_32gf algorithms have the highest FLOPs, and the regnetx_200mf and regnety_200mf algorithms have the lowest FLOPs. As the strength of the algorithms continued to increase, the FLOPs also continued to increase. The FLOPs of the SDS-Network algorithm in the regnet algorithm series were still very low. This further proves that the SDS-Network algorithm achieves a better lightweight effect.

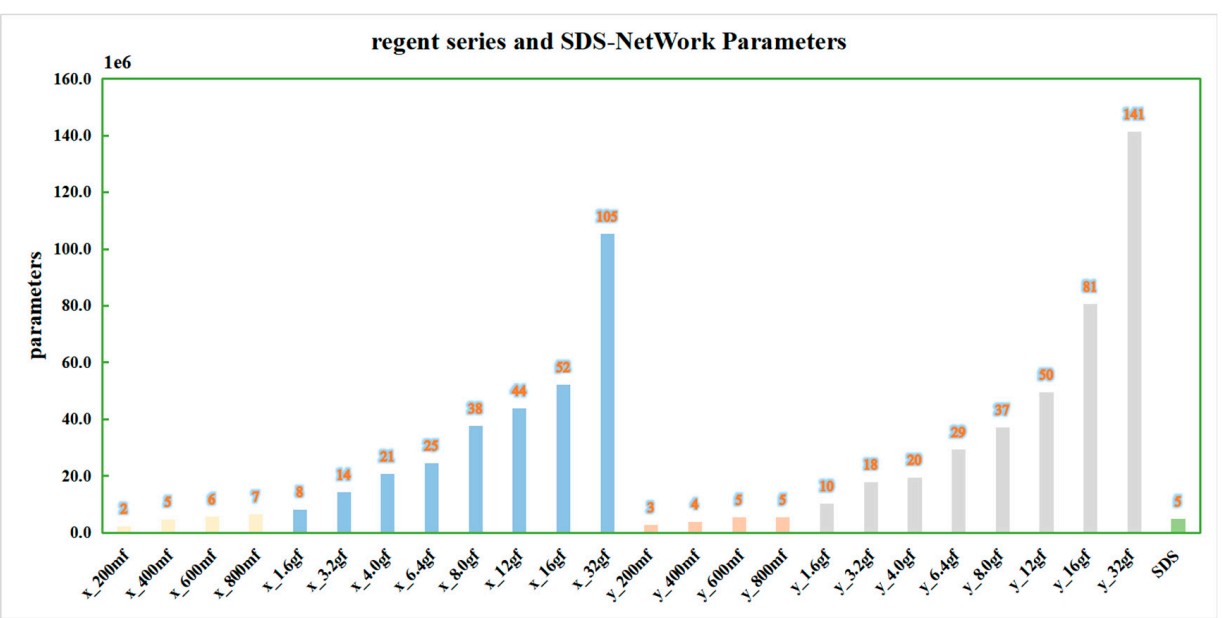

**Figure 11.** Parameter comparison of SDS-Network and regnet algorithm.

It can be seen in Figure 10 that, as the strength of the algorithm increased, the memory of the regnet series of algorithms also increased. Among them, the memory of the regnetx_32gf and regnety_32gf algorithms was still the highest, while that of the SDS-Network algorithm was the lowest at about 48.47 M.

It can be seen in Figure 11 that the change curve of parameters is similar to that of FLOPs and memory occupancy. The parameters of the SDS-Network algorithm in this paper are lower than those of the regnet series, and they are suitable for disaster classifications.

### 3.3. Class Activation Diagram

A Class Activation Map (CAM) [47] visualizes the process of algorithm recognition and presents it with an intuitive visual effect. In detail, red represents the recognized part, and blue represents the unattended part. This paper introduced Gradient-weighted Class Activation Mapping (Grad-CAM) [48], took the gradient of the feature map as the average, and obtained the N average gradient values corresponding to the N feature maps as the weight values. The information in the feature map can be used for discrimination when there is no observable difference. Compared with CAM, Grad-CAM can visualize the CNN of any structure without modifying the network structure or retraining, which further enhances the effect of algorithm recognition and visualization.

It can be seen in Table 8 that, in this paper, the final convolutional layer of the SDS-Network algorithm was classified and output by Grad-CAM to verify the quality of the algorithm. When determining the image category, the algorithm mainly uses the relevant part, which is displayed in red. Each category of the algorithm in this paper always has a red part, which proves that the algorithm in this paper recognizes an image by its characteristics rather than by accidental prediction. The darker the color of the picture, the stronger the attention of the model. It can be observed that there are about three attention points in each category in the no-background picture in Table 8. However, the blue is neglected, and the algorithm recognition effect is better; thus, it is suitable for disaster classifications.

**Table 8.** Grad-CAM data visualization table.

| SDS-Network | | | |
|---|---|---|---|
| *Background* | | **Category** | **Forecast Result** |
| *Yes* | No | | |
| | | Cyclone | True |
| | | Earthquake | True |
| | | Accident | False |
| | | Flood | True |
| | | Wildfire | True |

*3.4. Open Data Set*

**(1) Cifar-100 data set**

In order to further study the superiority of the algorithm, this paper classified the open data set Cifar-100 [49] into 100 categories, including 50,000 training sets and 10,000 test sets, which were iterated 50 times, 100 times, 150 times, and 200 times, and the Acc change graphs for the iterations of 100 times and 200 times were obtained.

In Table 9 and Figure 12, it can be seen that the algorithm reached equilibrium after 100 iterations, and after 200 iterations, the Acc of the algorithm was lower than that after 100 iterations. It can be stated that the algorithm has an overfitting problem after 200 iterations. Therefore, the algorithm can be iterated 150 times. After 50 iterations, the Acc of Top-1 reached 66.98, and that of Top-3 and Top-5 reached 83.85% and 88.97%, respectively, which indicates that the algorithm in this paper performed well and that it

had a faster effect. After 150 iterations, the Acc of Top-1, Top-3, and Top-5 reached 68.78%, 84.70%, and 89.75%, respectively, which proves that this algorithm was better and had a strong generalization ability.

**Table 9.** SDS-Network data sheet in Cifar-100.

| Model | FLOPs | Top-1 | Top-3 | Top-5 | Epoch |
|---|---|---|---|---|---|
| SDS-Network | 1.02 G | 0.6698 | 0.8385 | 0.8897 | 50 |
| | | 0.6790 | 0.8359 | 0.8891 | 100 |
| | | 0.6878 | 0.8470 | 0.8975 | 150 |
| | | 0.6928 | 0.8459 | 0.8954 | 200 |

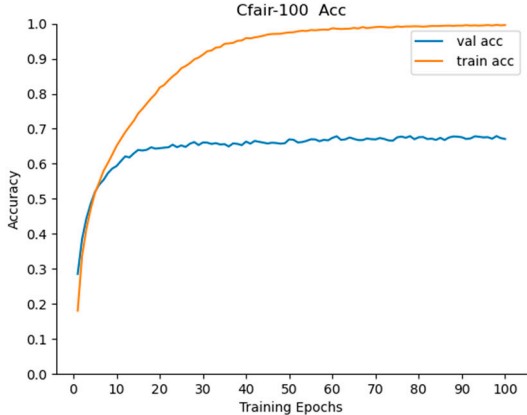

**Figure 12.** Acc change curve of 100 iterations of SDS-Network algorithm.

**(2) Caltech data set**

The Caltech pedestrian data set [50] used in this article has two categories: one is Caltech-101, and the other is Caltech-256. This article used Caltech-256, and each image had a size of 300 * 200, with 24, 581 images in the training set and 6026 images in the test set. It can be seen in Figure 13 that, when the SDS-Network algorithm was iterated 200 times, the model tended to be stable. It can be seen in Table 10 that, when the algorithm was iterated 200 times, the Acc of Top-1, Top-3, and Top-5 reached 53.50%, 67.24%, and 72.93%, respectively. We can therefore see that the Acc of Top-1 exceeds 50%, and after 50 iterations, it is close to 50%. In summary, the algorithm in this paper has a fast convergence speed, a high accuracy, and only 1.02 G FLOPs, so it is suitable for application in other classification tasks.

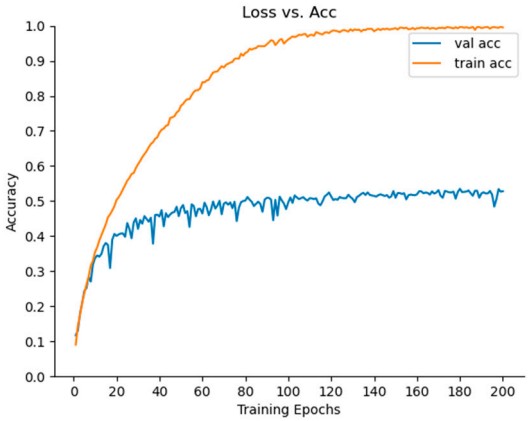

**Figure 13.** Acc change curve of SDS-Network algorithm iterated 200 times.

**Table 10.** SDS-Network data sheet in Caltech.

| Model | FLOPs | Top-1 | Top-3 | Top-5 | Epoch |
|---|---|---|---|---|---|
| SDS-Network | 1.02 G | 0.4839 | 0.6266 | 0.6965 | 50 |
| | | 0.5126 | 0.6434 | 0.7041 | 100 |
| | | 0.5285 | 0.6565 | 0.7149 | 150 |
| | | 0.5350 | 0.6724 | 0.7293 | 200 |

## 4. Conclusions

In recent years, natural disasters have occurred more frequently. In view of this, a disaster classification model was proposed in this paper to solve the low accuracy of current classification models. This article first used the spatial attention mechanism on ResNet to improve the accuracy of the algorithm, then used the Depthwise Separable Convolution to reduce the calculations and parameters of the algorithm, and finally used the hyperparameter fine-tuning method to adjust the model. This paper achieved good results when comparing the model proposed in this paper with classic classification models, and the algorithm proposed in this paper with lightweight algorithms and other newer algorithms. After that, the Grad-CAM visualization method was used to verify the correctness of the model's recognition, and then the data were published. It was found according to the experiments on Cifar-100 and Caltech that the performance of the algorithm in this paper is still good, which greatly verifies its generalization and portability.

**Author Contributions:** J.Y.: experiments and methodology; X.M.: funding and methodology; G.H.: methodology; S.L.: methodology; W.G.: methodology. All authors have read and agreed to the published version of the manuscript.

**Funding:** This work was supported by the National Natural Science Foundation of China (Grant No. 42171464, 41971283, 41801261, 41827801, 41801282), the Open Research Fund of National Earth Observation Data Center (grant number NODAOP2021005), and LIESMARS Special Research Funding.

**Informed Consent Statement:** Informed consent was obtained from all subjects involved in the study.

**Data Availability Statement:** The data used to support the findings of this study are included within the article. W. Xu, W. Wang, N. Wang, and B. Chen, "A New Algorithm for Himawari-8 Aerosol Optical Depth Retrieval by Integrating Regional PM2.5 Concentrations," in IEEE Transactions on Geoscience and Remote Sensing, doi:10.1109/TGRS.2022.3155503.

**Conflicts of Interest:** The authors state that they have no conflict of interest.

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
