# Peer review of "Research on Lightweight Disaster Classification Based on High-Resolution Remote Sensing Images"

_remotesensing, doi:10.3390/rs14112577_

Round 1

Reviewer 1 Report

The authors proposed ResNet in combination with spatial attention mechanism for classifying high-resolution RS imagery. They have compared their results with several other existing models in the related literature.

In my opinion, this paper addresses an interesting topic and offers some innovation. However, the topic presented here is a bit far from the topics we generally publish in MDPI RS journal; it is more an image processing related paper.

Abstract needs to be one paragraph. You do not need to separate objective method etc… in the abstract

The manuscript structure does not follow the MDPI RS structure. Introduction, method, etc… It starts with introduction then related work which I suppose it is background then it does to experimental link.

1.1 Remote sensing images section is not required in this case. The authors used a normal camera imagery. The content of this section is related to advanced RS apps for instance crop classification that normally carried out on real RS datasets.

In the same section the RS definition is wrong.

Please check the attached file for the minor comments.

Author Response

Dear Reviewer:

Thank you for your valuable review suggestions. The team will revise according to your review comments. We have made a word file and placed the revised content in the document below. thanks

Best Wishes: good health and smooth work
All members of the team

Reviewer 2 Report

Page 1, Abstract:

»proposes the SDS-Network algorithm«

You first mention the SDS-Network algorithm, but you did not explain prior what acronym “SDS” actually means.

Page 1, Abstract:

“the amount of model calculations and parameters”

You repeatedly use several times the same sentence in Abstract. It is sufficient to provide such information only once.

Page 2, Introduction:

“Nepal and central Italy;Scientists” => “Nepal and central Italy; Scientists”

Page 3, Section 2, Related work

I do not understand this section. Do you describe the related works, or do you describe your novel method. It is confusing. Your method should be described in the separate section. You should emphasize more clearly what is novelty in your approach.

Page 3, Subsection 2.1

“of H * H * C size, ”

What is “C”?

Page 4, Subsection 2.2:

“are shown in Formula 2, Formula 3, Formula 4, and Formula 5” => “are shown in Formulas 2-5”

Page 4, Subsection 2.2:

“can be seen from Formula 6 that the calculations of standard convolution are much greater than those of DC”

I do not understand how this can be seen from Formula 6!?

Page 5, Subsection 2.3:

“ the weight attenuation was set to the default value;”

And what is the default value?

Page 6, Subsection 3.1:

“ the length, width, and size of images are not limited”

What is the length of images? Actually, each image has the size, which is actually the pair (width, height)?

“are randomly cropped to a size of 224×224”

How exactly is this cropping performed?

Page 11-13, Subsection 3.2, Figures 8-11:

The diagrams (graphs) should be represented more clearly, so that the differences between the SDS-NetWork and regnet algorithm are more clearly visible (for example, the same X-axis, and different colors for each algorithm data).

Author Response

Dear Reviewer:

Thank you for your valuable review suggestions. The team will revise according to your review comments. We have made a word file and placed the revised content in the document below. thanks

Wish: good health and smooth work
All members of the team

Round 2

Reviewer 2 Report

The diagrams (Figure 8-11) are still not represented clearly, you only distinguished the regnet and the SDS-NetWork by labeling the X-axis, but in the graph, you use the same color for both the regnet and the SDS-NetWork - it is hard to directly compare the result (for example,  x_200_mf with y_200_mf). It would be better if you have the X-axis labeled for example 200_mf and present the regnet and the SDS-NetWork with different colors.

Author Response

(The authors gave the same response as above.)
